# A Novel Method for Baroreflex Sensitivity Estimation Using Modulated Gaussian Filter

**DOI:** 10.3390/s22124618

**Published:** 2022-06-18

**Authors:** Tienhsiung Ku, Serge Ismael Zida, Latifa Nabila Harfiya, Yung-Hui Li, Yue-Der Lin

**Affiliations:** 1Department of Anesthesiology, Changhua Christian Hospital, Changhua 50051, Taiwan; 21203@cch.org.tw; 2Ph.D. Program of Electrical and Communications Engineering, Feng Chia University, Taichung 40724, Taiwan; p0766289@o365.fcu.edu.tw; 3Department of Computer Science and Information Engineering, National Central University, Taoyuan 32001, Taiwan; latifanaharfiya@gmail.com (L.N.H.); or yunghui.li@foxconn.com (Y.-H.L.); 4AI Research Center, Hon Hai (Foxconn) Research Institute, Taipei 114699, Taiwan; 5Department of Automatic Control Engineering, Feng Chia University, Taichung 40724, Taiwan

**Keywords:** modulated Gaussian filter (modGauss), *α* index, Gaussian average filtering decomposition (GAFD), baroreflex sensitivity (BRS), autoregressive (AR), wavelet, Welch’s periodogram

## Abstract

The evaluation of baroreflex sensitivity (BRS) has proven to be critical for medical applications. The use of *α* indices by spectral methods has been the most popular approach to BRS estimation. Recently, an algorithm termed Gaussian average filtering decomposition (GAFD) has been proposed to serve the same purpose. GAFD adopts a three-layer tree structure similar to wavelet decomposition but is only constructed by Gaussian windows in different cutoff frequency. Its computation is more efficient than that of conventional spectral methods, and there is no need to specify any parameter. This research presents a novel approach, referred to as modulated Gaussian filter (modGauss) for BRS estimation. It has a more simplified structure than GAFD using only two bandpass filters of dedicated passbands, so that the three-level structure in GAFD is avoided. This strategy makes modGauss more efficient than GAFD in computation, while the advantages of GAFD are preserved. Both GAFD and modGauss are conducted extensively in the time domain, yet can achieve similar results to conventional spectral methods. In computational simulations, the EuroBavar dataset was used to assess the performance of the novel algorithm. The BRS values were calculated by four other methods (three spectral approaches and GAFD) for performance comparison. From a comparison using the Wilcoxon rank sum test, it was found that there was no statistically significant dissimilarity; instead, very good agreement using the intraclass correlation coefficient (ICC) was observed. The modGauss algorithm was also found to be the fastest in computation time and suitable for the long-term estimation of BRS. The novel algorithm, as described in this report, can be applied in medical equipment for real-time estimation of BRS in clinical settings.

## 1. Introduction

Baroreflex can be defined as a homeostatic process that controls blood pressure via the autonomic nervous system (ANS) negative feedback loop by modulating the total peripheral resistance and the heart rate (HR) [1]. The most common indicator used to assess the effect of baroreflex on heartbeat interval control is the baroreflex sensitivity (BRS) [2]. It has been suggested in the literature that BRS is a critical evaluation index in clinical settings. One study [3] showed that patients suffering from essential hypertension tended to have a lower BRS compared to normal subjects. A lower BRS was also found to be a direct indicator of the risk of sudden cardiac death after myocardial infarction [4]. In addition, BRS is considered to be a potential marker in the early detection of autonomic dysfunction in diabetic mellitus patients [5]. Patients with neurodegenerative disease are found to have declined BRS [6]. The estimation of BRS value has also been utilized for the diagnosis of brain stem death [7]. Furthermore, obstructive sleep apnea syndrome (OSAS) patients are found to have a significant decrease in BRS value during sleep. These observations are helpful in clarifying the cardiovascular pathophysiology of patients with OSAS [8]. A decreased BRS has also been observed in patients with acute intracerebral hemorrhage. This suggests that BRS could be a possible therapeutic marker for ischemic stroke [9]. The estimation of BRS is considered to be valuable for the clinical prognosis and diagnosis of various cardiovascular conditions, including heart failure and myocardial infarction [10]. It has been confirmed that BRS is a more reliable marker for the evaluation of functional status during the performance of physical and psychological tasks [11] compared to commonly used indices such as heart rate variability (HRV) [12], cardiac output (CO) [13] and blood pressure variability (BPV) [14]. BPV is a phenomenon characterizing the changes in blood pressure levels. It is the result of a rhythmical cardiac activity similar to HRV [15].

Baroreflex dysfunction has been shown to be strongly correlated with impaired cardiovascular performance, excess postoperative morbidity, and delays in hospital release for high-risk surgical patients [16]. Patients with chronic kidney disease (CKD) have been found to possess a reduced BRS and stiffer arteries, while peritoneal dialysis patients were observed to be worse than those in a non-dialysis group [17]. In addition, a decreased BRS is independently associated with post-stroke condition [18]. Growing evidence indicates that baroreflex pathways also project to key regions of the central nervous system (CNS) that regulate somatosensory and CNS arousal. These projections regulate pain perception, sleep, consciousness and cognition. This suggests that BRS estimation may be helpful in the evaluation of pain perception, sleep quality, consciousness level and cognitive function [19,20]. Only some representative, but not exhaustive, applications of BRS estimation in clinical settings are described above. 

Some conventional methods have been reported for BRS estimation, which include the neck chamber technique [21], injection of drugs [22,23], the Valsalva maneuver [24], or postural change [25] to stimulate systolic blood pressure (SBP) variations and monitor related variations on inter-beat interval (IBI). The above-mentioned approaches are subject to certain constraints in clinical settings. For the neck-chamber approach, the stimulus is limited to the baroreceptors in the carotid sinus. Regarding the drug-induced method, various drugs influence the baroreceptor in different ways and could have side-effects on different physiological systems. For certain patients, the Valsalva maneuver is not possible. Moreover, a dedicated test bed with a controllable tilt angle is usually required for the postural change approach such that the stimulus level in all experiments can be kept consistent. Several approaches have been suggested to overcome the above constraints. These approaches mainly focus on computing algorithms and can be categorized as spontaneous sequence methods (in the time domain) [26], spectral methods (in the frequency domain) [27,28] and cross-wavelet methods (in the time-frequency domain) [29]. Of these, the cross-wavelet approach is comparatively novel and can track BRS changes in the time as well as the frequency domains. However, it has the highest computing workload. Although the spontaneous sequence methods are the most commonly used, it is important to note that this methodology has shown some inherent bias in the estimation of BRS estimation, even during the resting state [30,31]. With respect to spectral methods, these enable the extraction of information about the sympathetic and parasympathetic nervous system activities individually [32]. This type of approach has been implemented in multiple commercial software packages, including Ponemah^®^ Software (Data Sciences International, St. Paul, USA), nevrokard^®^ BRS (Nevrokard Kiauta, Izola, Slovenia) and HemoLab Software (Harald Stauss Scientific, Dunuggan, USA). 

Recently, our team proposed a method termed Gaussian averaging filtering decomposition (GAFD) [33] to evaluate BRS, whereby the computation is conducted wholly in the time domain, but the results obtained are consistent with those obtained via conventional spectral techniques. GAFD is based on a three-layer tree structure that is similar to wavelet decomposition but only includes the Gaussian windows in different cutoff frequency. Its computation is more efficient than for conventional spectral methods, and, most importantly, there is no need to specify any parameter. The present paper introduces a new method, called modulated Gaussian filter (abbreviated as modGauss in this article) for BRS estimation. It is constructed with a more simplified structure than GAFD using only two bandpass filters of dedicated passbands to separately decompose the evenly resampled SBP and IBI sequences into two dedicated bands. This novel strategy makes modGauss more efficient than GAFD in the computation, yet the advantages of GAFD are preserved. 

The article is divided into the following sections: The proposed modGauss algorithm for BRS estimation is introduced in Section 2. This section also discusses the materials, conventional frequency-domain approaches, the GAFD algorithm for BRS estimation (in brief) and the statistical experiments performed in this study. The results of the analysis are provided in Section 3 along with corresponding discussion presented in Section 4. The conclusions of the article are summarized in Section 5.

## 2. Methods and Materials

This section is divided into five subsections. In the first subsection, the proposed algorithm is introduced. The following subsection briefly describes conventional spectral methods for BRS estimation. The estimation of BRS by the GAFD algorithm is concisely described in the third subsection. The fourth subsection describes the materials (including the signal resource and the computing environment) that were used for the computational experiments in this paper. The statistical testing methodologies used for comparing our developed algorithm with other methods are described in the final subsection.

### 2.1. BRS Estimation by modGauss 

The bandpass filters for the proposed method are constructed from the amplitude modulation on the Gaussian window. The Gaussian window of length $\left(2M+1\right)$ centralized at m=0 is set as follows [34]
(1)$$w\left[m\right]=\frac{\beta }{M\cdot \sqrt{2\pi }}e^{-\frac{1}{2}{\left(\beta \cdot \frac{m}{M}\right)}^{2}},\;for\;-M\leq m\leq M,$$
in which β stands for the scalar inversely related to the standard deviation of the Gaussian distribution. Under the condition β>2.5 [34], the truncated points of the Gaussian window are small enough to be omitted, and the associated discrete Fourier transform (DFT) is, in approximation, equal to
(2)$$W\left(\omega \right)\approx e^{-\frac{1}{2}\cdot {\left(\frac{M}{\beta }\cdot \omega \right)}^{2}},\;for\;-\pi <\omega \leq \pi .$$

From Equation (2), the spectrum of the Gaussian window substantially represents a non-causal lowpass filter (LPF) centered at ω=0 rad with no phase offset. To ensure that the energy remains unchanged, the Gaussian window have to be normalized as follows.
(3)$$w_{a}\left[m\right]=\frac{w\left[m\right]}{\sum_{i=-M}^{M}w\left[i\right]},\;for\;-M\leq m\leq M,$$
where $w\left[\cdot \right]$ is specified in Equation (1). One important issue of concerned with respect to the Gaussian window is the selection of β. Assume the −6 dB cutoff (where the decay is one half of the highest amplitude) is adopted and the −6 dB cutoff frequencies are located at $\omega =\pm \omega_{d}$ rad, then we have $W\left(\pm \omega_{d}\right)=0.5$ in Equation (2). By some algebraic manipulations, we have
(4)$$\beta =\frac{M\cdot \omega_{d}}{\sqrt{2\cdot \ln\left(2\right)}},\;$$
where $\omega_{d}=2\pi \cdot f_{d}/f_{s}$, $f_{d}$ is the −6 dB cutoff frequency and $f_{s}$ is the sampling frequency of the signal to be analyzed. Both $f_{d}$ and $f_{s}$ are in units of Hz. The bandpass filter in modGauss is acquired from the modulated Gaussian window $w_{a}\left[\cdot \right]$. Assume the resulting spectrum retains almost the same shape as the unmodulated spectrum. Let the carrier frequency in amplitude modulation be denoted as $f_{c}$ (in unit Hz), then the carrier signal can be generated by
(5)$$c_{r}\left[m\right]=\cos\left(2\pi \cdot \frac{f_{c}}{f_{s}}\cdot m\right),\;for\;-M\leq m\leq M.$$

The modGauss filter can then be derived as
(6)$$w_{g}\left[m\right]=w_{a}\left[m\right]\cdot c_{r}\left[m\right],\;for\;-M\leq m\leq M.\;$$

The spectrum of $w_{g}\left[\cdot \right]$ is the shifted version of $W\left(\omega \right)$ which is rescaled in amplitude and is now centered around $\omega =+2\pi \cdot f_{c}/f_{s}$ and $-2\pi \cdot f_{c}/f_{s}$ rad. If the scale β is selected properly, the split spectra will have no evident overlap. In this case, the generated $w_{g}\left[\cdot \right]$ shown in Equation (6) is, in essence, a non-causal bandpass filter with the passband ranging from $f_{c}-f_{d}$ to $f_{c}+f_{d}$ Hz. 

Assume the signal to be analyzed is denoted as $x\left[n\right]$, with 0≤n≤N−1. Because the modGauss filter is non-causal, the analyzed signal $x\left[\cdot \right]$ requires to be extended outside the endpoints. In this study, the extension is performed by the following formula
(7)$$x_{e}\left[n\right]=\left\{\begin{array}{c} 2\cdot x\left[0\right]-x\left[-n\right],\;for-M\leq n\leq -1\\ x\left[n\right],\;for\;0\leq n\leq N-1\;\\ 2\cdot x\left[N-1\right]-x\left[2\cdot N-n-2\right],\;for\;N\leq n\leq N+M-1 \end{array}\right..$$

After the signal extension, the filtering operation is conducted as follows
(8)$$y\left[n\right]=2\cdot \sum_{m=-M}^{M}w_{g}\left[m\right]\cdot x_{e}\left[m+n\right],\;for\;0\leq n\leq N-1.$$

The derived $y\left[\cdot \right]$ is the bandpass-filtered components buried in signal $x\left[\cdot \right]$. 

Another issue of concern for the modGauss filter is the determination of passband frequency range. In the spectral methods for human BRS estimation, the commonly adopted low-frequency (LF) band is within 0.04–0.15 Hz and the high-frequency (HF) band is located in the range 0.15–0.4 Hz [32]. In this condition, the center of the dedicated frequency range is selected to be the carrier frequency $f_{c}$, which is 0.095 Hz for the LF band and 0.275 Hz for the HF band, respectively. The deviated frequency from the center was previously denoted as $f_{d}$. It is assumed that the −6 dB cutoff frequencies are located at the endpoints of both bands in modGauss filters. In this case, the value of $f_{d}$ is equal to the deviation from $f_{c}$ to the end frequency. Therefore, it is 0.055 Hz for the LF band and 0.125 Hz for the HF band. With selection of the $f_{c}$ and $f_{d}$ values referred to, the proposed method is constructed from two modGauss filters, one for the LF band and the other for the HF band. The filtering operation is based on Equation (8), which is conducted fully in the time domain but can derive the frequency-domain information in the same way as for other spectral methods. 

The final issue for the modGauss filter requiring consideration is the filter length, which is determined by the parameter M. Because the components buried in the raw signal are closely related to the number of extremes (including both the local maxima and the local minima) appearing in the signal, the proposed method adopts an approach suggested by [35] according to the following formula
(9)$$M=2\cdot \lfloor \kappa \cdot \frac{N}{N_{e}}\rfloor ,$$
where κ is a scalar parameter ranging from 1.1 to 3, $N_{e}$ denotes the number of extremes, N represents the signal length and ⌊⋅⌋ refers to the operator rounding the value to the closest integer toward −∞. To select the feasible value of κ, the estimation of the BRS value is performed with different κ values (1,1, 2 and 3); the results are shown in Figure 1. It can be observed that the results are identical for each of the values of κ. The value 2 is in the middle of the selection range and is an integer, which means it can save storage space and offer fast computation in embedded applications. For these reasons, a constant value of 2 is selected for the parameter κ.

In addition, the second derivative test [36] is chosen for detecting the extremes. That is, the points whose first derivative is zero, and for which the second derivative is less than or greater than zero, are categorized as the local extremes. To prevent obtaining an implausible value of M, the limit of the extreme numbers is defined using the equation below
(10)$$N_{e_{TH}}=\lceil \frac{2\cdot \kappa \cdot N}{\frac{N}{2}-1}\rceil ,\;$$
in which ⌈⋅⌉ refers to the operator, rounding the value to the closest integer toward +∞. Signal processing is performed when the number of extremes detected $N_{e}$ is greater than $N_{e_{TH}}$. 

To facilitate readers’ understanding of the proposed method, the modGauss filter generation procedure for both the LF and HF bands is illustrated in Figure 2 from the time domain perspective. 

In this figure, the generation of both modGauss filters is based on Equations (1), (3)–(6) with M=50, $f_{s}=4$ Hz, $f_{c}=0.095$ Hz (for the LF band) and 0.275 Hz (for the HF band), and $f_{d}=0.055$ Hz (for the LF band) and 0.125 Hz (for the HF band). The corresponding modGauss filters from the frequency-domain perspective are shown in Figure 3, where the parameters are the same as those for Figure 2. It can be observed that the generated modGauss filters are bell-shaped bandpass filters centered at 0.095 and 0.275 Hz with the −6 dB cutoff frequencies corresponding to the frequency ranges of the LF and HF bands, respectively. 

As shown in Figure 3, it can be observed that both the modGauss filters are symmetrical with a center at index 0, and the computation of Equation (8) is inherently the convolution sum of $w_{g}\left[\cdot \right]$ and $x_{e}\left[\cdot \right]$. From the frequency-domain perspective, this is a direct multiplication of the frequency response of $w_{g}\left[\cdot \right]$ and the spectrum of $x_{e}\left[\cdot \right]$, and only the components within the dedicated frequency ranges are reserved. 

In this study, the SBP sequence expressed in mmHg, and its respective IBI sequence expressed in ms, are resampled using a 4 Hz sampling rate by cubic spline interpolation for BRS estimation. This is the primary reason why the parameter $f_{s}$ is set to 4 Hz in Figure 2 and Figure 3. Let the separated LF and HF component of the interpolated SBP sequence be designated as $x_{SBP,LF}\left[n\right]$ and $x_{SBP,HF}\left[n\right]$, for 0≤n≤N−1. The LF and HF band energies of SBP sequence are derived by
(11)$$\left\{\begin{array}{c} E_{SBP,LF}=\sum_{n=0}^{N-1}x_{SBP,LF}^{2}\left[n\right]\\ E_{SBP,HF}=\sum_{n=0}^{N-1}x_{SBP,HF}^{2}\left[n\right] \end{array}\right..$$

Let $x_{IBI,LF}\left[n\right]$ and $x_{IBI,HF}\left[n\right]$, with 0≤n≤N−1, denote the extracted LF and HF components for the interpolated IBI sequence. The LF and HF band energies for IBI sequence are acquired by
(12)$$\left\{\begin{array}{c} E_{IBI,LF}=\sum_{n=0}^{N-1}x_{IBI,LF}^{2}\left[n\right]\\ E_{IBI,HF}=\sum_{n=0}^{N-1}x_{IBI,HF}^{2}\left[n\right] \end{array}\right..$$

The BRS is then estimated using the so-called α index [37] in the LF band and the HF band, as well as in the averaged form, by
(13)$$\left\{\begin{array}{c} \alpha_{LF}=\sqrt{\frac{E_{IBI,LF}}{E_{SBP,LF}}}\\ \alpha_{HF}=\sqrt{\frac{E_{IBI,HF}}{E_{SBP,HF}}}\\ \alpha =\frac{\alpha_{LF}+\alpha_{HF}}{2} \end{array}\right.,\;$$
where the expression unit of all indices is ms/mmHg.

The computation procedure of our method is outlined in the following steps.

Step 1: The SBP sequence, expressed in mmHg, as well as its respective IBI sequence expressed in ms, are evenly resampled by $f_{s}$ Hz, which is a 4-Hz cubic spline interpolation in this study. 

Step 2: Find the number of extremes for both the interpolated sequences. Then, determine the parameters M and β according to Equation (9) and Equation (4), respectively. In this study, the second derivative test [36] is applied for detection of extremes. In addition, the value of $f_{d}$ is 0.055 Hz for the LF band and 0.125 Hz for the HF band. If the number of extremes is less than $N_{e_{TH}}$, which is derived according to Equation (10), this implies the interpolated sequence is nearly a monotonic trend wave and the energies in both the LF and HF bands can be considered to be zero. The BRS estimation is only conducted once the number of detected extremes is greater than $N_{e_{TH}}$. 

Step 3: Generate the carrier signals for both the LF and HF bands according to Equation (5), where the value of $f_{c}$ is 0.095 Hz for the LF band and for the HF band is 0.275 Hz. 

Step 4: Extend the interpolated SBP and IBI sequences beyond the endpoints according to Equation (7), and then conduct the filtering process according to Equation (8) for both the LF and HF bands separately. 

Step 5: Derive the energies of the interpolated SBP and IBI sequences in both the LF and HF bands according to Equation (11) and Equation (12), respectively. Finally, the BRS values are estimated according to Equation (13).

### 2.2. BRS Estimation with Spectral Methods

Three conventional spectral approaches are adopted for performance comparison in this study. One is the Welch’s periodogram [38], whereas the others are the autoregressive (AR) spectrum estimation [39] and the wavelet [40]. The SBP, as well as the IBI sequences, should be resampled uniformly by $f_{s}$ Hz, which is a 4-Hz cubic spline interpolation in this study, before conducting the spectral analysis. 

For the Welch’s periodogram, the parameters of the 512-point Hanning window and 512-point fast Fourier transform (FFT), along with 50% overlap among adjacent windows, are selected for performance comparison. The energies of the LF band and the HF band are deduced by direct addition of the periodograms in the respective frequency range. Let the derived energies in LF and HF bands for the interpolated SBP and IBI sequences be represented by $E_{SBP,LF}$, $E_{SBP,HF}$, $E_{IBI,LF}$ and $E_{IBI,HF}$, respectively. The BRS values are then estimated by Equation (13). 

Another commonly used method for spectrum analysis is the AR model [39]. It consists of an all-pole approach whose transfer function is given by
(14)$$H\left(z\right)=\frac{\sigma }{1+\sum_{i=1}^{P}a_{i}\cdot z^{-i}}=\frac{\sigma }{\prod_{i=1}^{P}\left(1-p_{i}\cdot z^{-1}\right)’}.$$
where σ is the scalar related to the estimation error, $a_{i}$ (1 ≤i≤P) are the AR coefficients and $p_{i}$ (1 ≤i≤P) denote the poles of the model. The order of the AR model is represented by the parameter P. The AR model of fixed order 25 is selected for both the IBI sequences and an interpolated SBP, and Burg’s maximum extropy method is used to derive the estimation error σ as well as the AR coefficients. This method has the advantage of high resolution, even for short data records, and model stability [41]. The poles of the model, $p_{i}$ (1≤i≤P), can be derived directly by factorization after the AR coefficients $a_{i}$ (1 ≤i≤P) have been acquired. The resulting spectrum obtained from the acquired AR model is defined as
(15)$$P_{AR}\left(f\right)=\frac{\sigma^{2}}{{\left|\prod_{i=1}^{P}\left(1-p_{i}\cdot e^{-j\cdot 2\pi \cdot \frac{f}{f_{s}}}\right)\right|}^{2}}.$$

Let $P_{d}\left(f_{k}\right)$ denote the energy contributed from the dominant pole $p_{k}$ (the pole closest to the unit circle) corresponding to the frequency $f_{k}$ (in unit Hz) for a certain band of the AR spectrum, then $P_{d}\left(f_{k}\right)$ can be derived as follow [42].
(16)$$P_{d}\left(f_{k}\right)=\frac{\sigma^{2}}{\left|p_{k}\cdot \prod_{i=1,i\neq k}^{P}\left(p_{k}-e^{j2\pi \cdot \frac{f_{i}}{f_{s}}}\right)\cdot \prod_{i=1}^{P}\left(\frac{1}{p_{k}}-e^{-j2\pi \cdot \frac{f_{i}}{f_{s}}}\right)\right|},$$
where $f_{i}$ represents the corresponding frequency (in unit Hz) of the pole $p_{i}$ (1≤i≤P). The values of $P_{d}\left(f_{k}\right)$ are estimated from the dominant poles appearing within the HF band (0.15–0.4 Hz) and the LF band (0.04–0.15 Hz) for the IBI and SBP spectrum. The acquired $P_{d}\left(f_{k}\right)$ values for the SBP and IBI spectra from the LF and HF bands are used to represent the energies $E_{SBP,LF}$, $E_{SBP,HF}$, $E_{IBI,LF}$ and $E_{IBI,HF}$ for BRS estimation. In a similar way to the approach using Welch’s periodogram, the BRS values are estimated according to Equation (13). 

The advantage of spectral analysis by the scalogram of complex Morlet wavelet is the ability to focus on analysis within the dedicated frequency range with a selected frequency resolution [40]. In this study, the frequency resolution is selected to be 0.005 Hz. After having acquired the scalogram under pre-determined scales (which are the scaled reciprocal of frequencies) in different time segment, the averaged spectrum is then time-averaged along each frequency. The energy within the HF band (0.15–0.4 Hz) and the LF band (0.04–0.15 Hz) for the SBP and IBI can then be estimated by Equation (13). 

### 2.3. BRS Estimation by GAFD

GAFD [33] is a novel approach for BRS estimation which adopts a three-level structure similar to wavelet decomposition but using Gaussian lowpass filters of three different cutoff frequencies (which are 0.04, 0.15 and 0.4 Hz, respectively). The SBP sequence in mmHg and its associated IBI sequences in ms were resampled evenly in 4 Hz by cubic spline interpolation. Before the signal decomposition, the Gaussian lowpass filters for GAFD are based on Equation (3), and the signal must also be extended out of the endpoints according to Equation (7). The parameters M and β are determined by Equations (9) and (4), with the values of $f_{d}$ being 0.04, 0.15, and 0.4 Hz, respectively, for GAFD. The filtering operation is conducted by Equation (8), with the filter $w_{g}\left[\cdot \right]$ being replaced by $w_{a}\left[\cdot \right],$ as shown in Equation (3). In the following content, the symbol $x\left[\cdot \right]$ is used to represent the resampled SBP or IBI sequences. The procedure for BRS estimation by GAFD is briefly summarized below. A detailed introduction is provided in [33]. 

Step 1: Perform the lowpass filtering to divide the sequences $x\left[\cdot \right]$ to components of above 0.4 Hz and below 0.4 Hz. Let the components below 0.4 Hz be referred to as $x_{04}\left[\cdot \right]$. 

Step 2: Conduct the lowpass filtering on $x_{04}\left[\cdot \right]$ to divide the sequences to components above 0.15 Hz and below 0.15 Hz. Let the components below 0.15 Hz be referred to as $x_{015}\left[\cdot \right]$. The components above 0.15 Hz represent the HF components for the estimation of BRS and are designated as $x_{HF}\left[\cdot \right]$. Let the HF components for IBI and SBP sequences be denoted by $x_{IBI,HF}\left[\cdot \right]$ and $x_{SBP,HF}\left[\cdot \right]$. The HF band energies can therefore be calculated by squared summation, as shown in Equations (11) and (12). 

Step 3: Perform the lowpass filtering on $x_{015}\left[\cdot \right]$ to divide the sequences to components above 0.04 Hz and below 0.04 Hz. The components below 0.04 Hz represent the LF components for the estimation of BRS and are designated as $x_{LF}\left[\cdot \right]$. Let the LF components for IBI and SBP sequences be expressed by $x_{IBI,LF}\left[\cdot \right]$ and $x_{SBP,LF}\left[\cdot \right]$. In a similar way to the HF band, the LF band energies can also be calculated by squared summation, as shown in Equations (11) and (12). 

Step 4: Lastly, the values of BRS are approximated with the α index in the LF band and the HF band, along with the averaged form, independently, as shown in Equation (13).

The MATLAB codes for the BRS estimation by spectral approaches and GAFD referred to in this article can be found on the website: https://figshare.com/articles/software/Sample_codes_for_the_assessment_of_baroreflex_sensitivity_BRS_/12820430 (accessed on 10 June 2022).

### 2.4. Materials 

For the purpose of this research, both beat-by-beat SBP and IBI sequences were obtained from the EuroBavar dataset [43]. The dataset, which has been used in previous studies [33,44,45] to compare the performance of different methods used for BRS estimation, can be downloaded online. The EuroBavar dataset contains 46 recording files acquired from 21 subjects composed of 4 males and 17 females. The participating subjects are organized as follow:12 normotensive outpatients, including two patients who were treated for hypercholesterolemia, one diabetic patient with no cardiac neuropathy and one woman who was three months pregnant,two patients who were treated for hypertension,one non-treated hypertensive patient,two patients suffering from cardiac autonomic failure (one with heart transplantation and one patient with diabetic neuropathy)four healthy volunteers.

ECG signals (Cardiocap™ II, Datex-Ohmeda, Helsinki, Finland) and continuous non-invasive blood pressure (Finapres^®^, Finapres Medical Systems, Enschede, The Netherlands) were recorded on all subjects at a sampling frequency of 500 Hz by 16-bit analog-to-digital conversion resolution in the standing position (files marked with the penultimate letter “S”) and supine position (files marked with the penultimate letter “L”) in a silent room according to a uniform procedure. Informed consent was obtained for all participants prior to collecting the EuroBavar dataset. Furthermore, the ECG and blood pressure signals from this dataset are available as waveforms (files with the ending letter “C”) and beat-by-beat sequences (files with the ending letter “B”). For all the computer experiments of the study, the beat-by-beat SBP, along with the IBI sequences, were adopted. 

All the statistical analyses and computational experiments in this study were performed on a MacBook Pro (mid 2015, with 2.5 GHz Intel Core i7 and 16 GB 1600 MHz DDR3 memory, Apple Inc., Cupertino, CA, USA) using MATLAB^®^ (2020b, MathWorks, Inc., Natick, MA, USA).

### 2.5. Statistical Analysis 

The values of the BRS values were derived from Equation (13) by Welch’s periodogram, then by GAFD and finally by modGauss. Various statistical tests were applied to assess the effectiveness of the suggested modGauss method. Firstly, the Shapiro–Wilk normality test [46] was performed to check if the BRS values in the whole dataset, as well as in the dataset for the supine and standing positions, were normally distributed or not. We then used non-parametric statistical methods for testing; the Wilcoxon signed-rank test [47] was used to assess if there was a statistical difference in the BRS values between the standing and supine positions for each approach. The Wilcoxon rank sum test [48] was employed to assess whether any statistical difference existed between the modGauss method and the other four methodologies. Lastly, to assess the level of correlation between modGauss and the others methodologies, the intraclass correlation coefficient (ICC) [49,50] was determined. A MATLAB algorithm, developed by Salarian [51] and based on [49], was adopted for the estimation of the 95% confident intervals (CI) and the ICC. Six different types of ICC computation can be used for the ICC analysis; however, in the context of this study, the ICC for a two-way mixed approach and single score (type “A-1” in the code) was chosen for the computation of the ICC; the lower and upper bounds of the ICC were calculated assuming a significance level of 0.05.

## 3. Results

### 3.1. Experimental Results in Time Domain and Frequency Domain

This article proposes a new method called modGauss for the estimation of BRS using α indices according to Equation (13). Figure 2 and Figure 3 have previously demonstrated the generation procedure of modGauss filters from time- and frequency-domain perspectives, respectively. To demonstrate the effect of the modGauss filters, the decomposed results for one data record in the EuroBavar dataset (the record numbered B002LB) are shown in Figure 4 and Figure 5 (for the resampled SBP sequence); Figure 4 presents the results from a time-domain perspective, while Figure 5 shows the corresponding results from a frequency-domain perspective.

The decomposed results for one data record of the EuroBavar dataset (B002LB) are presented in Figure 6 and Figure 7 (for the resampled IBI sequence). Figure 6 shows the results in the time domain, while Figure 7 describes the corresponding results in the frequency domain.

The Welch’s periodogram (with 512-point FFT, 512-point Hanning window and a 50% overlap between the neighboring windows) was used to acquire the spectra for the resampled sequences and the decomposed signals in Figure 5 and Figure 7. For the figures in the time domain, it can be observed that the raw signals were separated into the lower-frequency components and the higher-frequency components (refer to Figure 4 and Figure 6). However, it is very evident that the spectra of the lower-frequency and higher-frequency components have been separated to the discrete ranges of 0.04–0.15 Hz and 0.15–0.4 Hz by the modGauss filters from the corresponding frequency-domain perspective (as shown in Figure 5 and Figure 7).

### 3.2. Performance Evaluation of the modGauss

To evaluate the performance and effectiveness of our proposed method, four other approaches—GAFD, AR spectrum, Welch’s periodogram and wavelet—were chosen for performance comparison. The present study used the EuroBavar dataset to confirm the applicability of the newly suggested approach. Although this dataset has been used in other studies [33,44,45], it is important to note that few studies have shown all the derived BRS values for each data record in the dataset. Figure 8 represents the BRS value for each of the methods discussed in this study. It can be observed that the median BRS value was almost identical for each of the five methods, which indicates that the proposed modGauss algorithm is comparable to the other methodologies for BRS estimation. 

After estimation of the BRS values for each of the methods, detailed statistical values were obtained using different tests, including the Shapiro–Wilk normality test, the Wilcoxon signed-rank test, the Wilcoxon rank sum test and the intraclass correlation coefficient (ICC). First, the Shapiro–Wilk normal test [46] was applied to assess if the BRS values for the whole dataset, as well as for the datasets for supine and standing position, were normally distributed. In Table 1 below, the test results for the five methods, including modGauss, GAFD, AR, Welch and Wavelet in different conditions (whole dataset, supine and standing position) are presented. 

It was observed that all the *p* values were smaller than 0.05, indicating that there was no Gaussian distribution of BRS values for each test. Based on this, the non-parametric Wilcoxon signed-rank test [47] was utilized to assess if there was a statistical difference in the BRS values between the standing and supine positions for each of the four methods. The experimental results for this test are shown in Table 2. 

Since all the *p* values were lower than 0.05, this implies that there were statistical differences in the BRS values for the supine and standing positions for all the five approaches. Therefore, a further test was needed to determine if there was any statistically significant difference between the proposed modGauss and the other four methods for BRS estimation. The Wilcoxon rank sum test [48] was then to this end; Table 3 shows the comparison results under each condition (supine and standing position, whole dataset). 

It was observed that all the *p* values were higher than 0.05, which implies that there was no substantial difference between the modGauss and the other four BRS estimation approaches. The ICC [49,50] was used to evaluate the level of correlation between the BRS values deriving from modGauss and the other four methodologies. The results of the ICC analysis comparing modGauss with the other approaches under different conditions (supine and standing position, whole dataset) are shown in Table 4. The 95% confidence interval (CI) is also provided in this table for each comparison, along with the ICC value. Even considering the lowest CI (which is 0.91 for $\alpha_{LF}$ compared between the modGauss and GAFD under the supine condition), the obtained results indicated “excellent” agreement (based on the guideline described in [52]) for the estimation of BRS between the modGauss and the other four approaches.

## 4. Discussion

The various experiments and tests performed using the proposed modGauss algorithm have shown that the algorithm is highly comparable to other conventional BRS estimation methods. More importantly, a major advantage of our proposed modGauss method is its simplicity for performing complex computation. As its highly simplified structure uses only two dedicated bandpass filters and bypasses the three-level structure of the GAFD algorithm [33], this strategy is more computationally efficient than the GAFD method while retaining all the advantages of the latter. Another major advantage of the proposed modGauss method is its fast computation time. A fast-computational time is an important criterion in defining the performance of computational algorithms. In this study, a comparative analysis of the proposed modGauss method was performed to evaluate the computational time of the algorithm using the EuroBavar dataset. Four different methods (GAFD, AR, Welch and Wavelet) were chosen for comparison and the results are presented in Table 5. It can be observed from this table that the modGauss algorithm had a shorter computational time in the 0.05 quartile, median and 0.95 quartile. The *p* values of the Wilcoxon rank sum one-tailed tests for modGauss versus the other four methods were all less than 0.05. This means that the modGauss algorithm was faster in computation, which makes it more attractive for performing rapid computation for BRS estimation. Although the elapsed time for each method was of the order of milli-seconds, it should be noted that the values were evaluated under the platform of a mid-2015 MacBook Pro. If the computation were conducted in the embedded micro-controller of a medical device, the elapsed time might be very different, especially for the spectral methods because of their relatively high demand for computational resources. 

When selecting a computational algorithm for BRS estimation, a key consideration is the ability of the algorithm to perform long-term analysis. In this regard, a comparative analysis was performed for long-term BRS estimation between the proposed modGauss and sleep stage, with the results shown in Figure 9. The signal slp45 from the MIT-BIH Polysomnographic Database [53] of PhysioNet [54] was adopted for the long-term BRS analysis. In this figure, the proposed modGauss algorithm was used to perform an analysis of the BRS estimate for a duration of 6 h. From Figure 9, it can be seen that the BRS values generally increased after the awake stages, which implies arousal of the baroreflex function when a person is awoken from the sleep stage. With respect to sleep, the increase in BRS values and the delay in BRS elevation after waking up are topics which may warrant further study. 

Moreover, when compared to the other four methods reviewed in this study, the long-term analysis of the modGauss algorithm suggest that it performed as well as the other methods, as shown in Figure 10. This, combined with the previous observations on elapsed time (Table 5), leads to the conclusion that, overall, the modGauss algorithm is better and more suitable for the long-term estimation of BRS. 

Although the new modGauss method proposed in this paper has various advantages, limitations may include the double reflection that is required at the start and end points of the filter to perform signal filtering. In practical use, a short waiting time is required to assure the signal entirely entering the modGauss filtering window. However, if the results obtained in this waiting window can be omitted without influencing the clinical decision, this limitation does not affect the performance of the algorithm because the modGauss algorithm only uses convolution, which results in faster computation time, as shown in Table 5. The methodologies, such as AR, Welch, and wavelet require more complex computation for model estimation, complex multiplication or to be spread out in the time and frequency domains, which causes them to have longer computation times because of the relatively higher demand on computing resources. The novel modGauss algorithm as discussed in this paper, provides new insights into computational methods for BRS estimation. Future development of this work could lead to the incorporation of the modGauss algorithm into embedded medical devices for real-time BRS analysis. Real-time estimation of BRS is important in clinical application as it provides critical information for continuous control of patient blood pressure [55,56]. Despite the existence of other methods for real-time BRS estimation [57,58], these methods typically use advanced microcontrollers for computation, making them more demanding on resources and more expensive. The modGauss, which is a based on a linear convolutional approach, does not require significant computational resources or advanced microcontroller systems. 

For BRS estimation, the phase spectrum is sometimes adopted in spectral approaches to depict the timing difference from the sympathetic and vagal nerves in different conditions [29]. In addition, some spectral approaches utilize the threshold of coherence (e.g., greater than 0.5) between SBP and IBI spectral estimates to determine the bands used for the computation of BRS [45]. The lack of phase and coherence information is another limitation of the presented modGauss algorithm because its computation is conducted thoroughly in the time domain. For this issue, correlation analysis between the separated SBP and IBI sequences in the LF and HF bands may be helpful to compensate for such constraints. Due to the differences in the selected band and in the spectrum analysis method, there have been eleven different spectral approaches for BRS estimation proposed [45]. Because there has been no gold standard method for BRS estimation until now, this study selected three representative spectral analysis methods for performance comparison. 

BRS estimation has been shown to be crucial in clinical settings and the evaluation of BRS has been applied in cardiovascular disease (CVD) [3,4,9,10,18], diabetes [5], neurodegenerative disease [6], diagnosis of brain stem death [7], OSAS [8], surgery [16], physiological function evaluation while conducting physical and psychological tasks [11], CKD [17], and pain perception [19,20]. No dedicated medical device has been available for BRS estimation until now. One attractive approach is to embed the BRS estimation algorithm in contemporary medical equipment. Accumulated observations from BRS estimation in various situations may provide new insights into the role that baroreflex function plays in clinical settings. The proposed algorithm is a strong candidate for practical implementation because of its computational simplicity and efficiency.

## 5. Conclusions

A new algorithm named modGauss, developed for the estimation of BRS values, is described in this paper. The proposed method utilizes two bandpass filters of dedicated passbands to separately decompose the evenly resampled SBP and IBI sequences into LF and HF bands. Convolution and squared summation are used without the need to specify a parameter, thus rendering the developed algorithm highly efficient in computing. The EuroBavar dataset was used to perform computational experiments and the BRS estimation results were compared with four different methods, namely the Welch’s periodogram, GAFD, the AR spectrum and wavelet. It was found that the BRS values under different conditions (whole dataset, supine position and standing position) obtained through modGauss showed “excellent” concordance compared to the other four methodologies. In addition, further tests showed that the proposed modGauss algorithm, which has a very simplified structure, had faster computation times compared with the other methodologies (Table 5) and was also suitable for the long-term analysis of SRB estimation (Figure 9 and Figure 10). The proposed algorithm may open new research avenues for the development of faster embedded medical devices for clinical applications.

## Figures and Tables

**Figure 1 sensors-22-04618-f001:**
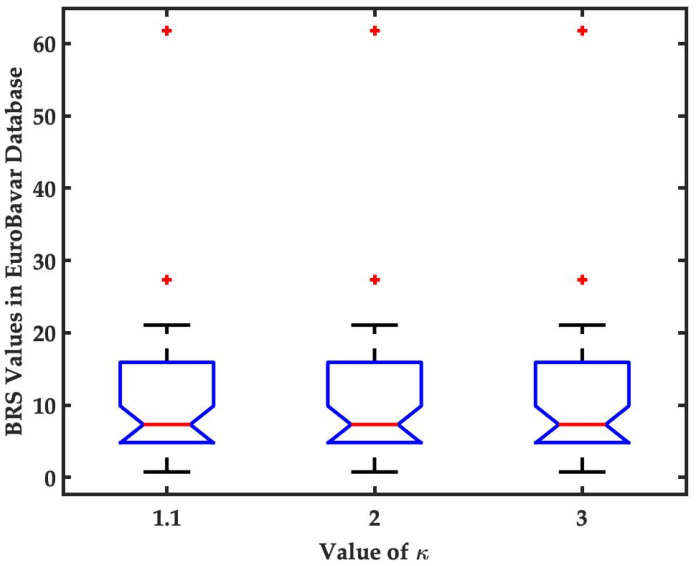
BRS analysis with different values of κ (1.1, 2 and 3). It can be observed that the BRS value is identical for each value of κ. In this study, κ is constantly set to 2. The data points beyond the whiskers are marked by red **+** symbol.

**Figure 2 sensors-22-04618-f002:**
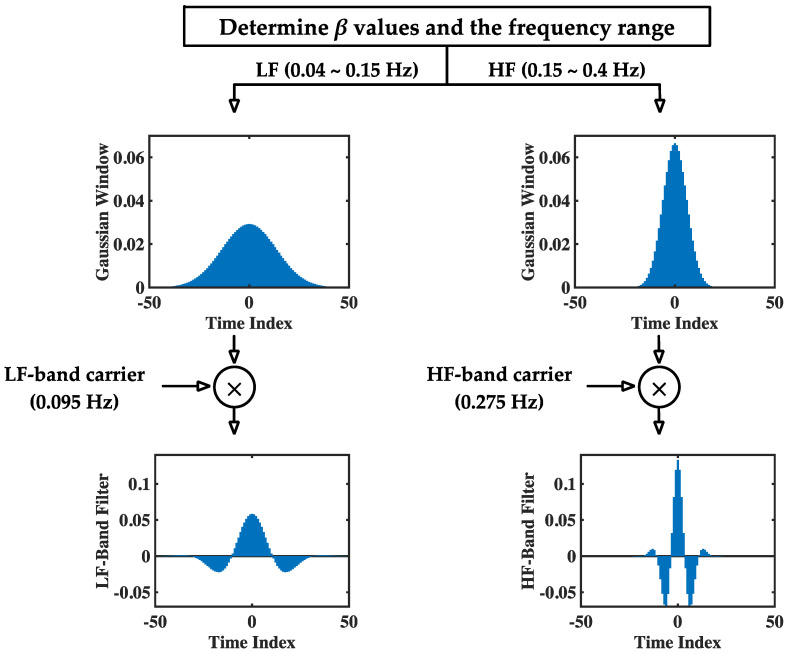
Generation procedure for the modGauss filters in LF and HF bands (from time-domain perspective).

**Figure 3 sensors-22-04618-f003:**
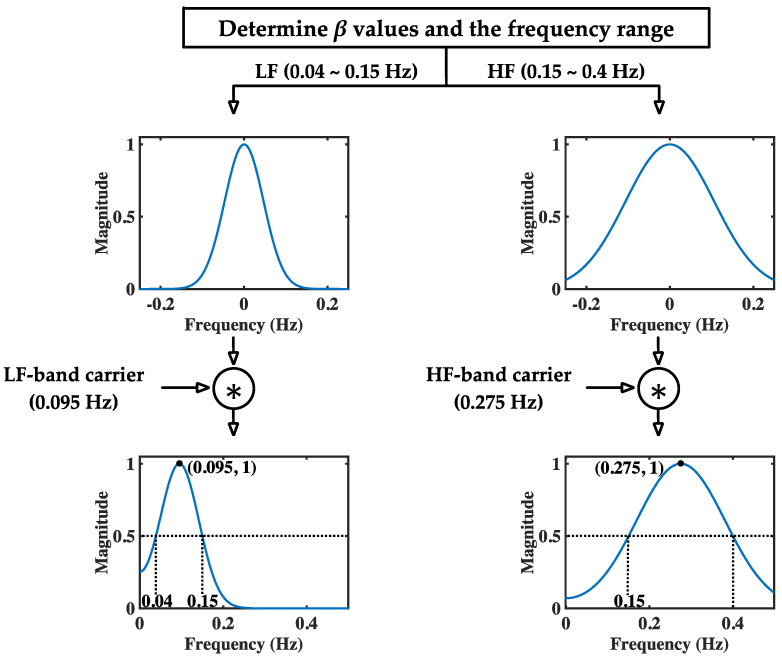
Generation procedure for the modGauss filters in LF and HF bands (from frequency-domain perspective).

**Figure 4 sensors-22-04618-f004:**
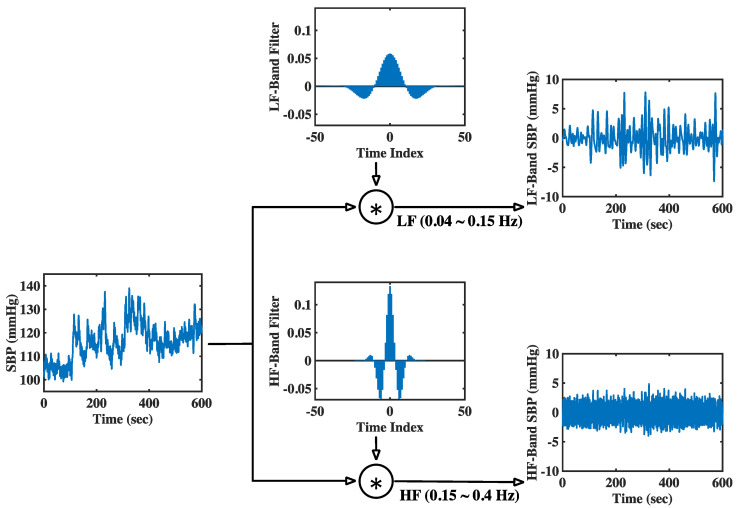
Derived results for the interpolated SBP sequence in both the LF and HF band. Results from time-domain perspective with the B002LB data of EuroBavar dataset as example.

**Figure 5 sensors-22-04618-f005:**
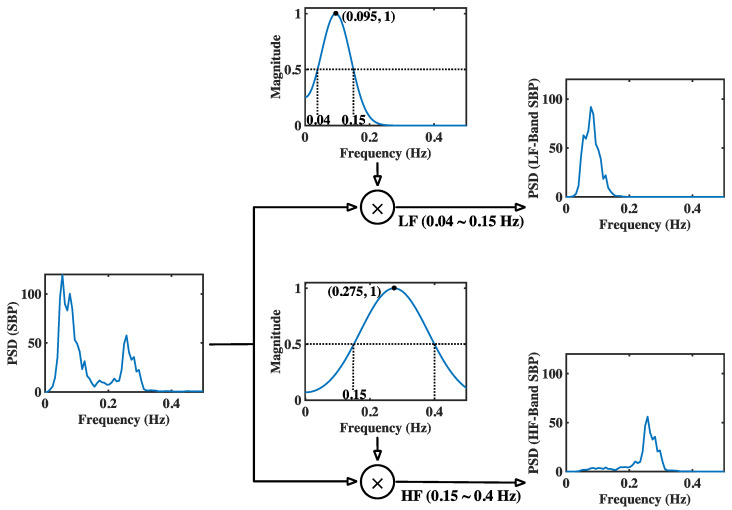
Derived results for the interpolated SBP sequence in both the LF and HF band. Results from frequency-domain perspective with the B002LB data of EuroBavar dataset as example.

**Figure 6 sensors-22-04618-f006:**
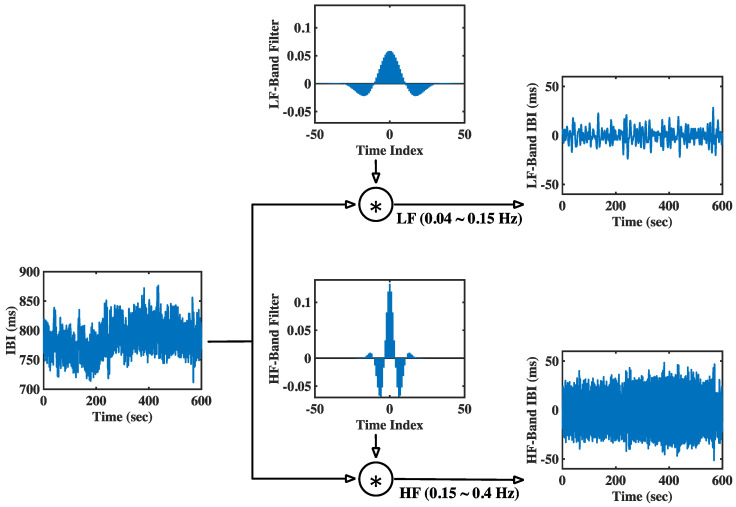
Derived results for the interpolated IBI sequence in LF band and HF band (from time-domain perspective, with the data record B002LB of the EuroBavar dataset as an example).

**Figure 7 sensors-22-04618-f007:**
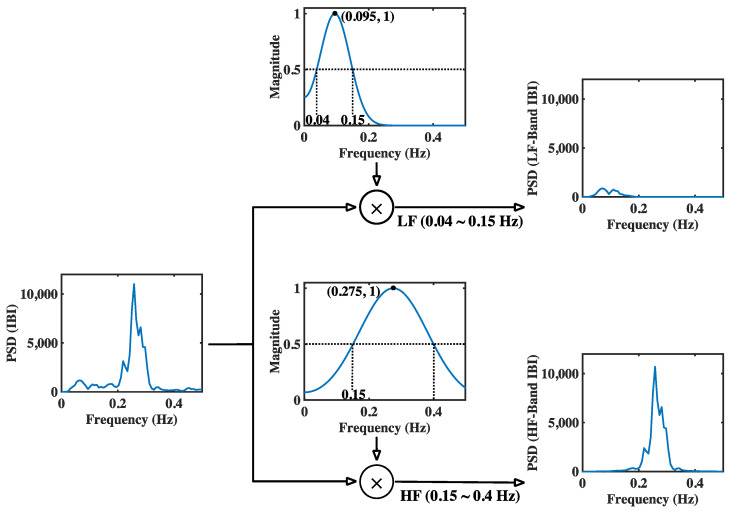
Derived results for the interpolated IBI sequence in LF band and HF band (from frequency-domain perspective, with the data record B002LB of the EuroBavar dataset as an example).

**Figure 8 sensors-22-04618-f008:**
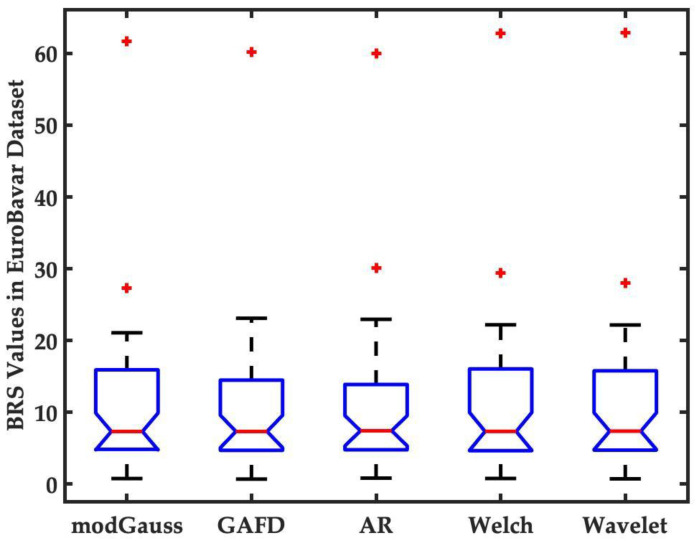
Estimation of the BRS values by five different methods including modGauss, GAFD, AR spectrum, Welch’s periodogram and wavelet. The data points beyond the whiskers are marked by red **+** symbol.

**Figure 9 sensors-22-04618-f009:**
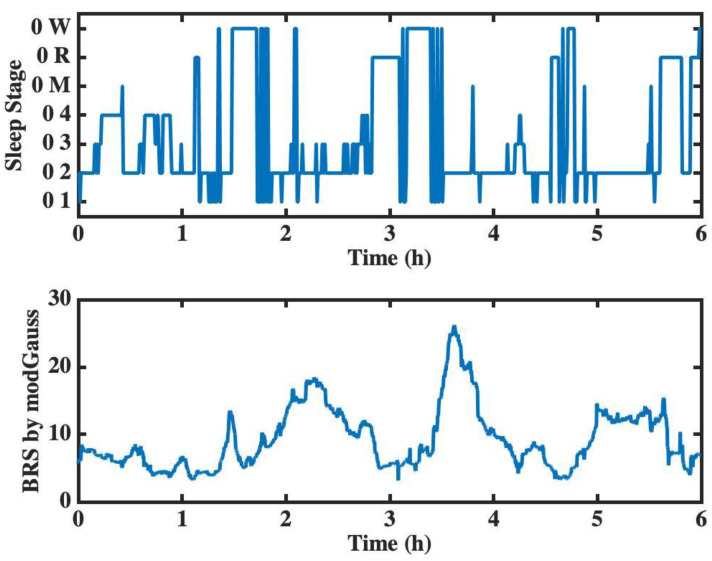
Long-term BRS analysis by modGauss in comparison with sleep stage (with the sleep stages 1–4 being denoted by 01–04, the subject motion by 0M, rapid eye movement sleep by 0R, and the awake stage by 0W).

**Figure 10 sensors-22-04618-f010:**
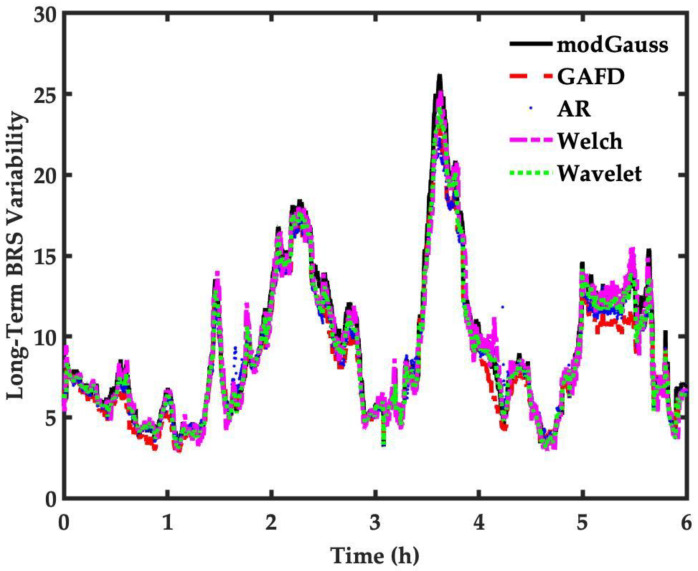
Long-term sleep BRS analysis by five methods including GAFD, AR spectrum, Welch’s periodogram and Wavelet.

**Table 1 sensors-22-04618-t001:** Summary of the test results for each of the five methods following the Shapiro–Wilk normality test.

Condition	modGauss	GAFD	AR	Welch	Wavelet
α (Whole)	$6.52\times 10^{-8}$	$4.81\times 10^{-8}$	$1.65\times 10^{-7}$	$1.49\times 10^{-7}$	$6.01\times 10^{-8}$
α (Supine)	$3.37\times 10^{-5}$	$2.13\times 10^{-5}$	$8.18\times 10^{-5}$	$7.51\times 10^{-5}$	$3.38\times 10^{-5}$
α (Standing)	$5.30\times 10^{-3}$	$1.03\times 10^{-2}$	$2.35\times 10^{-2}$	$5.55\times 10^{-3}$	$7.01\times 10^{-3}$
$\alpha_{LF}$ (Whole)	$7.95\times 10^{-9}$	$7.28\times 10^{-9}$	$6.85\times 10^{-9}$	$9.05\times 10^{-9}$	$3.24\times 10^{-10}$
$\alpha_{LF}$ (Supine)	$2.77\times 10^{-6}$	$1.97\times 10^{-6}$	$2.21\times 10^{-6}$	$2.82\times 10^{-6}$	$3.25\times 10^{-7}$
$\alpha_{LF}$ (Standing)	$1.12\times 10^{-2}$	$1.60\times 10^{-2}$	$9.84\times 10^{-3}$	$1.26\times 10^{-2}$	$1.58\times 10^{-2}$
$\alpha_{HF}$ (Whole)	$8.85\times 10^{-7}$	$2.99\times 10^{-7}$	$1.18\times 10^{-6}$	$1.15\times 10^{-6}$	$7.77\times 10^{-6}$
$\alpha_{HF}$ (Supine)	$3.58\times 10^{-4}$	$2.11\times 10^{-4}$	$9.86\times 10^{-4}$	$8.33\times 10^{-4}$	$4.71\times 10^{-3}$
$\alpha_{HF}$ (Standing)	$2.10\times 10^{-3}$	$3.13\times 10^{-3}$	$1.19\times 10^{-2}$	$1.92\times 10^{-3}$	$1.56\times 10^{-3}$

Legend: modGauss: using modulated Gaussian filter; GAFD: using Gaussian average filtering decomposition; AR: by autoregressive (AR) power spectral density (PSD); Welch: using Welch’s periodogram; Wavelet: by the scalogram using complex Morlet wavelet; α (Test Condition): the *p* value of BRS α index under different “Test Condition”; $\alpha_{LF}$ (Test Condition): the *p* value of BRS α index in LF band under different “Test Condition”; $\alpha_{HF}$ (Test Condition): the *p* value of BRS α index in HF band under different “Test Condition”; Test Condition: “Whole” refers to the analysis of the entire dataset; “Supine” stands for the analysis of the data in supine position; “Standing” stands for the analysis of the data in standing position.

**Table 2 sensors-22-04618-t002:** Summary of the test results for each of the five methods following the Wilcoxon signed-rank test.

Index	modGauss	GAFD	AR	Welch	Wavelet
α	$3.52\times 10^{-5}$	$4.02\times 10^{-5}$	$6.75\times 10^{-5}$	$3.52\times 10^{-5}$	$3.08\times 10^{-5}$
$\alpha_{LF}$	$2.06\times 10^{-4}$	$1.73\times 10^{-3}$	$5.88\times 10^{-4}$	$2.06\times 10^{-4}$	$3.72\times 10^{-4}$
$\alpha_{HF}$	$4.02\times 10^{-5}$	$5.22\times 10^{-5}$	$1.43\times 10^{-4}$	$5.94\times 10^{-5}$	$3.52\times 10^{-5}$

Legend: modGauss: using modulated Gaussian filter; GAFD: using Gaussian average filtering decomposition; AR: by autoregressive (AR) power spectral density (PSD); Welch: using Welch’s periodogram; Wavelet: by the scalogram using complex Morlet wavelet; *α*: represents the BRS *α* index. The BRS α index *p* values for supine versus standing cases are reported on the same row in each method. $\alpha_{LF}$: represents the BRS *α* index in LF band. The BRS α index *p* values in LF band for supine versus standing cases are reported on the same row in each method. $\alpha_{HF}$: represents the BRS *α* index in HF band. The BRS α index *p* values in HF band for supine versus standing cases reported on the same row in each method.

**Table 3 sensors-22-04618-t003:** Wilcoxon rank sum test results for the proposed modGauss compared to different methods.

Condition	vs. GAFD	vs. AR	vs. Welch	vs. Wavelet
α (Whole)	0.73	0.90	0.84	0.89
α (Supine)	0.60	0.74	0.79	0.88
α (Standing)	0.96	0.91	1.00	0.93
$\alpha_{LF}$ (Whole)	0.49	0.70	0.80	0.73
$\alpha_{LF}$ (Supine)	0.42	0.64	0.91	0.74
$\alpha_{LF}$ (Standing)	0.78	0.84	0.74	0.73
$\alpha_{HF}$ (Whole)	0.88	0.86	0.81	0.81
$\alpha_{HF}$ (Supine)	0.78	0.79	0.71	0.86
$\alpha_{HF}$ (Standing)	0.93	0.98	0.93	0.81

Legend: modGauss: by modulated Gaussian filter; GAFD: using Gaussian average filtering decomposition; AR: by autoregressive (AR) power spectral density (PSD); Welch: using Welch’s periodogram; Wavelet: by the scalogram using complex Morlet wavelet; α (Test Condition): the *p* value of BRS α index under different “Test Condition”; $\alpha_{LF}$ (Test Condition): the *p* value of BRS α index in LF band under different “Test Condition”; $\alpha_{HF}$ (Test Condition): the *p* value of BRS α index in HF band under different “Test Condition”; Test Condition: “Whole” refers to the analysis of the entire dataset; “Supine” stands for the analysis of the data in supine position; “Standing” stands for the analysis of the data in standing position.

**Table 4 sensors-22-04618-t004:** Summary of the intraclass correlation coefficient (ICC) results for modGauss compared to four different methodologies.

Condition	vs. GAFD		vs. AR		vs. Welch		vs. Wavelet	
α (Whole) 95% CI	1.00		0.99		1.00		1.00	
	0.99	1.00	0.99	1.00	0.99	1.00	0.99	1.00
α (Supine) 95% CI	0.99		0.99		1.00		1.00	
	0.96	1.00	0.98	1.00	0.99	1.00	0.99	1.00
α (Standing) 95% CI	1.00		0.99		1.00		1.00	
	0.99	1.00	0.98	1.00	0.99	1.00	0.99	1.00
$\alpha_{LF}$ (Whole) 95% CI	0.99		1.00		1.00		1.00	
	0.96	1.00	0.99	1.00	0.99	1.00	0.99	1.00
$\alpha_{LF}$ (Supine) 95% CI	0.99		1.00		1.00		1.00	
	0.91	1.00	0.98	1.00	0.99	1.00	0.99	1.00
$\alpha_{LF}$ (Standing) 95% CI	0.99		1.00		1.00		1.00	
	0.99	1.00	0.99	1.00	0.99	1.00	0.99	1.00
$\alpha_{HF}$ (Whole) 95% CI	1.00		0.98		0.99		0.99	
	0.99	1.00	0.97	0.99	0.98	1.00	0.99	1.00
$\alpha_{HF}$ (Supine) 95% CI	1.00		0.98		0.99		0.99	
	0.99	1.00	0.96	0.99	0.96	1.00	0.98	1.00
$\alpha_{HF}$ (Standing) 95% CI	1.00		0.97		0.99		0.99	
	0.99	1.00	0.94	0.99	0.98	1.00	0.98	1.00

Legend: The data presentation in the table above follows the format of ICC (in upper cell) and the 95% CI of ICC (in lower cell), with CI designating the confidence interval. modGauss: using modulated Gaussian filter; GAFD: using Gaussian average filtering decomposition; AR: by autoregressive (AR) power spectral density (PSD); Welch: using Welch’s periodogram; Wavelet: by the scalogram using complex Morlet wavelet; α (Test Condition): the test results of BRS α index under different “Test Condition”; $\alpha_{LF}$ (Test Condition): the test results of BRS α index in LF band under different “Test Condition”; $\alpha_{HF}$ (Test Condition): the test results of BRS α index in HF band under different “Test Condition”; Test Condition: “Whole” refers to the analysis of the entire dataset; “Supine” stands for the analysis of the data in supine position; “Standing” stands for the analysis of the data in standing position.

**Table 5 sensors-22-04618-t005:** Summary of the elapsed time tests using the EuroBavar dataset for five different methods.

**Method**		**modGauss**	**GAFD**	**AR**	**Welch**	**Wavelet**
p value (SW)		$4.88\times 10^{-11}$	$3.21\times 10^{-10}$	$3.22\times 10^{-11}$	$7.46\times 10^{-12}$	$4.10\times 10^{-11}$
Elapse time (s)	0.05 quartile	0.0093	0.0117	0.0129	0.0155	0.0704
	Median	0.0109	0.0128	0.0135	0.0162	0.0746
	0.95 quartile	0.0159	0.0166	0.0347	0.0345	0.1524
p value (Wilcoxon)		1.0	$2.89\times 10^{-10}$	$1.59\times 10^{-12}$	$1.55\times 10^{-14}$	$7.41\times 10^{-17}$

Legend: modGauss: by modulated Gaussian filter; GAFD: using Gaussian average filtering decomposition; AR: by autoregressive (AR) power spectral density (PSD); Welch: using Welch’s periodogram; Wavelet: by scalogram using complex Morlet wavelet; SW denotes the Shapiro–Wilk normal test (0.05 significance level); Wilcoxon represents the Wilcoxon rank sum one-tailed test (0.05 significance level).

## Data Availability

Not applicable.

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
