# Peer review of "A Novel Method for Baroreflex Sensitivity Estimation Using Modulated Gaussian Filter"

_sensors, 2022, doi:10.3390/s22124618_

Round 1
Reviewer 1 Report
After reading the revised version of the article, I have no objections to the entire work. The authors in the revised version took into account the comments of the editor and reviewers, so that the shortcomings of the first version of the work were removed. The new version of the paper is satisfactory.
Author Response
Dear Reviewer
We want to express our acknowledgement to your affirmation as well as the valuable comments and the time you have spent on our article. We have addressed the replies for all reviewers. Because we just received the comments from three reviewers on 26 May, 2022, based on the comments given in your informed Email at the same day, only the replies for three reviewers are shown below.
Thank you for your affirmation. Your previous comments have
improved our article to better quality. We would like to thank you for the valuable comments as well as the time which you have spent on this article.
Reviewer 2 Report
It looks weird/unprofessional when you use gmail addresses, but that’s up to the editor to point out.
You main claim here is that, “This novel strategy makes mod-Gauss more efficient than GAFD in the computation, yet the advantages of GAFD are still preserved”. You mean computationally efficient, and yet I couldn’t find an analysis of this computational efficiency. Did I miss something? So how long do these calculations take?
Other than that my impression is that you’ve provided a great description and analysis of your methods, and the paper is well written. But it shouldn’t be published without more data on the computational efficiency of these methods compared.
Author Response
Dear Reviewer
We would like to thank you for the time in reviewing this article and for the valuable comments and remarks.In this revised version, we have corrected the Email address of Dr. Tienhsiung Ku to be 21203@cch.org.tw, which is Dr. Ku’s official Email address. We respect Mrs. Harfiya’s decision and keep her Email address unchanged. The reason is she will earn her Ph. D. degree soon from National Central University (NCU). We are not sure whether she can receive any Email via her Email address in NCU after her graduation. Please accept our apology on this issue.
For the comparison of computational efficiency, we have added the related analysis in the previous version according to the comments given by the academic editor and the other reviewer. The analysis result is summarized in Table 5 (on Page 19, Lines 547-555), and the related description is shown at Lines 537-546 (on Page 19). The related content has been marked in red color to make it easily distinguished from the original or added content, which is shown as follows.
“A fast-computational time is an important criterion in defining performant computational algorithm. In this work, a comparative analysis of the proposed modGauss method were performed to evaluate the computational time of the algorithm using the EuroBavar dataset. Four different methods (GAFD, AR, Welch and Wavelet) were chosen and the results are presented in Table 5. It can be observed from this table that the modGauss algorithm has a shorter computational time in 0.05 quartile, Median and 0.95 quartile. The p values of the Wilcoxon rank sum one-tailed tests for modGauss versus the other four methods are all less than 0.05. This means that the modGauss algorithm is faster in computation, which subsequently renders it more attractive to perform rapid computation for BRS estimation.”
Reviewer 3 Report
My primary objection is that I do not see a motivation for this work.
From the description of the method, it can be deduced that the authors proposed a Gaussian filter with a central frequency in the middle of the human physiological LF and HF ranges and a cutoff of 6dB at the edges of these ranges. Then they filter the physiological signals and estimate sBRS according to the known procedure for spectral estimation. They compare their method to four other methods, one of them their own.
I must admit that it was impossible to find anything new in this contribution.
Description of Gaussian filter in time and frequency domain and its illustrative application in Figures 4, 5, 6, and 7 are nice, but for a student’s tutorial that could be used in a Digital filters course, not for a scientific paper.
Estimation of sBRS by Eq. (13) is long known, and, besides, the authors did not comment on coherence or phase that are well-known thresholds used to modify equation (13).
The proposed method is compared to 4 other methods: one of their own, and three that perform PSD estimation in three different ways. The students in signal processing laboratories write reports on the comparison of much more different PSD estimation techniques.
The authors use the signals from the EuroBaVar study that has 420 quotations, but they fail to comment that this study compared 21 different sBRS estimation methods, out of which 11 were based on spectral analysis. The authors did not compare their results to the results of the EuroBaVar study, although they use the same signals.
I have noticed minor technical omissions, but, since my opinion is that this paper performs the tasks that are common at laboratories on signal processing courses, I shall not state them.
Author Response
Dear Reviewer
Thanks for your valuable comments and time having spent on this article. In this article, we try to show a novel approach for BRS estimation and to compare the results with the other four methods. For the PSD approaches, only three representative techniques are selected for comparison to make the demonstration concise and precise but not to blur the proposed idea presented in this article. To reply the issues you have mentioned above, we have added new content in the discussion (on Pages 21-22, Lines 595-606) to make the constraints of the presented algorithm more complete. The added content is shown as follows.
“For BRS estimation, the phase spectrum is sometimes adopted in spectral approaches to depict the timing difference from the sympathetic and vagal nerves in different conditions [29]. In addition, some spectral approaches utilize the threshold of coherence (e.g., greater than 0.5) between SBP and IBI spectral estimates to determine the bands used for the computation of BRS [45]. The lack of phase and coherence information is another limitation of the presented modGauss algorithm because its computation is thoroughly conducted in the time domain. For this issue, the correlation analysis between the separated SBP and IBI sequences in LF and HF bands may be helpful to compensate such constraints. Due to the differences in the selected band and in the spectrum analysis method, there have been eleven different spectral approaches for BRS estimation [45]. Because there is no gold standard method for BRS estimation till now, this article selected three representative spectral analysis methods for performance comparison.”
Round 2
Reviewer 2 Report
Thank you for addressing my comments, good job.
Author Response
We want to express our acknowledgement to your affirmation as well as the valuable comments and the time you have spent on our article. The article cannot be improved to its present quality without your kind help.This manuscript is a resubmission of an earlier submission. The following is a list of the peer review reports and author responses from that submission.
Round 1
Reviewer 1 Report
The abstract should be restructured and written with the primary objective, methodology, results, and conclusion.
It is not clear the main physiological contribution of this novel approach.
Please briefly introduce to Blood Pressure Variability analysis in the introduction section.
The presentation of the results section should be restructured. It is not clear the comparison about the methods. Kindly indicate if the tables show p-values or mean values.
Other authors have previously introduced a novel approach, a continuous wavelet transform-based processing for estimating the power spectrum content of heart rate variability during hemodiafiltration. Please check the computation of power in that article and verify if it could apply to your work.
The conclusion is too large, and it does not reflect the main contribution of this paper.
Reviewer 2 Report
A new approach namely modulated Gaussian filter for baroreflex sensitivity estimation is presented in the research. The experimental results of the proposed system have been compared with the existing methods. The authors claim that the work can be applied in medical equipment and be used for a real-time estimation of baroreflex sensitivity in hospitals. The article is well written however the results shown in tabulated form are too lengthy. In my opinion the graphical representation of the results would be more appealing for the readers.
Reviewer 3 Report
The paper concerns the problem of estimation of the baroreflex sensitivity which is crucial for determination of a homeostatic process of blood pressure controling. The most frequently used indicator to evaluate the effect of baroreceptors on heart rate interval control is baroreceptor sensitivity (BRS). And about the estimation of this index (alpha) is this paper.
This paper presents a new approach to BRS evaluation based on a modulated Gaussian filter. The authors emphasize that this approach provides a simpler structure than Gaussian average filtering decomposition (GAFD). The results obtained are satisfactory and comparable to the state-of-the-art methods.
The document in its current form is quite limited, and therefore authors must address the following recomendations:
The authors' approach does not impress me with this work. Since this paper is a modification of the method that was presented in [29] Y. D. Lin, S. I. Zida, "Estimation of baroreflex sensitivity by Gaussian average filtering decomposition", Biomed. Signal Process. Control, 68 (2021) 102576.
A thorough analysis of both works shows a significant similarity, e.g. in the number of formulas (in [29] there are 17, and there are 16 of this work), the number of tables in both works is identical (5 tables), in work [29] there are 41 items. There are 48 bibliographic items in the reviewed paper, the structure of the reviewed paper is also very similar. The thing that makes this work different from the work of [29] is the use of a modulated signal (equation (5)). There are also differences in the drawings that show the main idea of ​​the work. It is a pity that Figures 3 and 4 as well as 5 and 6 do not show the differences in the SBP and IBI sequences for the methods discussed in the paper, or whether there are, for example, differences in the obtained LF and HF spectra.
The authors write "An" excellent "concordance for BRS estimation was observed when comparing modGauss and the other three approaches, which was due to the lowest CI of ICC reaching 0.90 (occurred at the between the modGauss and GAFD comparison in the supine condition)" which is shown by the results obtained in tables, but what other advantages does the modGauss method have? Is it less computational effort? less computational complexity? or shorter computation time?
What are the limitations of the proposed method? What are they due to?
What are the possible further developments of the proposed method?
In relation (9) there is a kappa parameter and it is assumed that kappa=2 although it is indicated that kappa can belong to the interval <1.1, 3>. How much does the value of the kappa parameter affect the results obtained?